# PATE-GAN: Generating Synthetic Data with Differential Privacy Guarantees

**James Jordon**[*]
Engineering Science Department
University of Oxford, UK
`james.jordon@wolfson.ox.ac.uk`

**Jinsung Yoon**[*]
Department of Electrical and Computer Engineering
UCLA, California, USA
`jsyoon0823@g.ucla.edu`

**Mihaela van der Schaar**
University of Cambridge, UK
Department of Electrical and Computer Engineering, UCLA, California, USA
Alan Turing Institute, London, UK
`mihaela@ee.ucla.edu`

## Abstract

Machine learning has the potential to assist many communities in using the large datasets that are becoming more and more available. Unfortunately, much of that potential is not being realized because it would require sharing data in a way that compromises privacy. In this paper, we investigate a method for ensuring (differential) privacy of the generator of the Generative Adversarial Nets (GAN) framework. The resulting model can be used for generating synthetic data on which algorithms can be trained and validated, and on which competitions can be conducted, *without* compromising the privacy of the original dataset. Our method modifies the Private Aggregation of Teacher Ensembles (PATE) framework and applies it to GANs. Our modified framework (which we call PATE-GAN) allows us to tightly bound the influence of any individual sample on the model, resulting in tight differential privacy guarantees and thus an improved performance over models with the same guarantees. We also look at measuring the quality of synthetic data from a new angle; we assert that for the synthetic data to be useful for machine learning researchers, the relative performance of two algorithms (trained and tested) on the synthetic dataset should be the same as their relative performance (when trained and tested) on the original dataset. Our experiments, on various datasets, demonstrate that PATE-GAN consistently outperforms the state-of-the-art method with respect to this and other notions of synthetic data quality.

## 1 Introduction

More and more large datasets are becoming available in a wide variety of communities. In the U.S. medical community, for example, the fraction of providers using electronic health records (EHR) increased from 9.4% in 2008 to 83.8% in 2015 [20]. The availability of large datasets presents enormous opportunities for collaboration between the data-holders and the machine learning community. However, many of these large datasets, especially EHR, include sensitive information that prevents data-holders from sharing the data.

The most common way to mitigate the privacy risk of sharing sensitive records is to de-identify the records - but it is by now well-known that records that have been de-identified can be easily re-identified by linking them to other identifiable datasets [30; 13; 24; 22; 14]. (This is especially true for medical records of patients who have rare diseases.) However, if the purpose of sharing the data is to develop and validate machine learning methods for a particular task (e.g. prognostic risk scoring), real data is not necessary; it would suffice to have synthetic data that is *sufficiently like* the real data.

---

[*]: Equal contribution

Precisely what this means depends on how the synthetic data will be used. For example, the synthetic data may be used to train models that will be deployed directly on real data. In this setting it is important that these methods (which we trained entirely on synthetic data) perform as well as if they had been trained on real data. Another setting to consider is one in which data-holders wish to use the synthetic data to identify the best method(s) to be used on the real data [11]. In this setting, it is not important that training on synthetic data leads to good performance on real data, but rather that comparing two methods on the synthetic data results in conclusions similar to those that would have been drawn from comparing the two methods on the real data. We evaluate our method in both settings.

Generative Adversarial Networks (GAN) [19] provide a powerful method for using real data to generate synthetic data but it does not provide any rigorous privacy guarantees. Our method modifies the GAN machinery in a way that *does* guarantee privacy; the synthetic data is (differentially) private [12] with respect to the original data. To do this we modify the training procedure of the discriminator to be differentially private by using a modified version of the Private Aggregation of Teacher Ensembles (PATE) [25; 26] framework. The Post-Processing Theorem [12] then guarantees that the GAN generator - which is trained only using the differentially private discriminator - will also be differentially private and thus so will the synthetic data it generates. We call our proposed framework PATE-GAN.

Using two Kaggle datasets, two different real-world medical datasets and two UCI datasets, we evaluate the utility of the samples generated by PATE-GAN in various settings with various levels of differential privacy. In line with the settings outlined above, we consider two methods for evaluating the similarity of synthetic datasets with a real dataset. The first method, first proposed in [15], compares the predictive performance of models trained on the synthetic datasets and tested on the real dataset. The second method, which we propose for the first time here, compares the performance rankings of predictive models on the synthetic datasets with their performance rankings on the real dataset. We demonstrate that, for both of these methods, PATE-GAN consistently produces synthetic datasets that are "more like" the original real dataset than the synthetic datasets produced by the state-of-the-art benchmark (DPGAN [32]).

The contributions of this paper can be summarized as follows: (1) we modify the PATE framework and apply it to GANs to generate synthetic data, (2) we demonstrate in the experiments section that using PATE to enforce differential privacy results in higher quality synthetic data than DPGAN using various real-world datasets, (3) we propose a novel new metric for evaluating the generated synthetic data.

## 2 RELATED WORKS

The most related previous work to this paper is DPGAN [32]. Like us, DPGAN proposes a framework for modifying the GAN framework to be differentially private, also relying on the Post-Processing Theorem to change the problem of learning a differentially private generator to learning a differentially private discriminator. Their work uses a technique introduced by [1] that provides a differentially private mechanism for training deep networks. The key idea is that noise is added to the gradient of the discriminator during training to create differential privacy guarantees. These ideas are also used in [2]. Our method is similar in spirit; during training of the discriminator differentially private training data is used, which results in noisy gradients, however, we use the mechanism introduced in [25] which we believe gives tighter differential privacy guarantees (via tighter bounds on the effect of a single sample) than those provided in [1]. This means that for the same privacy guarantees, our method is capable of producing higher quality synthetic data. For a visual representation of both PATE-GAN and DPGAN, see the Appendix.

The proposed model modifies the PATE framework [25; 26] for use in a generative model setting (specifically for use with GANs). The key to the GAN framework is that the discriminator is a *differentiable* module trained to classify samples as either real or generated. The PATE framework provides a differentially private mechanism for classification by training multiple teacher models on disjoint partitions of the data. To classify a new sample each teacher's output is evaluated on the sample and then all outputs are noisily aggregated. This noisy aggregation, though, results in a classifier that is *not* differentiable with respect to the parameters of the generator. In order to overcome this problem we follow the idea of the student model, also proposed in [25], that

involves taking some *public* unlabelled data, labelling it using the standard PATE mechanism and then training the student using the resulting labelled data. Because access to any public data is often an unreasonable assumption in synthetic data generation, we adapt this training paradigm in a way that does not require public data by training the student using only outputs from the (differentially private) generator.

Some previous works generate synthetic data using summary statistics of the original data [23] or based on specific domain-knowledge [6]; however, those methods are limited to low-dimensional feature spaces, specific fields and do not provide any differential privacy guarantees. [9] generates synthetic patient records using a GAN framework. However, [9] focuses only on generating discrete variables, whereas PATE-GAN is capable of generating mixed-type (continuous, discrete, and binary) variables. Furthermore, [9] also does not provide any differential privacy guarantees and instead uses ad-hoc notions of privacy which are only validated empirically.

Finally, it is worth remarking that it is known to be hard in the worst-case to generate private synthetic data [31] and so techniques such as GANs are necessary to address this challenge.

## 3 BACKGROUND

Let us denote the feature space by $\mathcal{X}$, the label space by $\mathcal{Y}$ and write $\mathcal{U} = \mathcal{X} \times \mathcal{Y}$. Let the dimension of $\mathcal{U}$ be $d$. Suppose that $\mathbf{X}$ and $Y$ are random variables over $\mathcal{X}$ and $\mathcal{Y}$. We write $\mathbf{U} = (\mathbf{X}, Y)$ and $\mathbf{x}, y, \mathbf{u}$ to denote realizations of $\mathbf{X}, Y$ and $\mathbf{U}$, respectively. The dataset $\mathcal{D}$ consists of $N$ samples of $\mathbf{u}$, assumed i.i.d. according to $\mathcal{P}_U$ denoted as $\mathcal{D} = \{\mathbf{u}_i\}_{i=1}^{N} = \{(\mathbf{x}_i, y_i)\}_{i=1}^{N}$.

### 3.1 DIFFERENTIAL PRIVACY

We first provide some preliminaries on differential privacy [12] before describing PATE-GAN; we refer interested readers to [12] for a thorough exposition of differential privacy. We will denote an algorithm by $\mathcal{M}$, which takes as input a dataset $\mathcal{D}$ and outputs a value from some output space, $\mathcal{O}$.

**Definition 1.** *(Neighboring Datasets) Two datasets $\mathcal{D}, \mathcal{D}'$ are said to be neighboring if*

$$\exists x \in \mathcal{D} \ s.t. \ \mathcal{D} \setminus \{x\} = \mathcal{D}'. \tag{1}$$

**Definition 2.** *(Differential Privacy) A randomized algorithm, $\mathcal{M}$, is $(\epsilon, \delta)$-differentially private if for all $\mathcal{S} \subset \mathcal{O}$ and for all neighboring datasets $\mathcal{D}, \mathcal{D}'$:*

$$\mathbb{P}(\mathcal{M}(\mathcal{D}) \in \mathcal{S}) \leq e^\epsilon \mathbb{P}(\mathcal{M}(\mathcal{D}') \in \mathcal{S}) + \delta \tag{2}$$

where $\mathbb{P}$ is taken with respect to the randomness of $\mathcal{M}$.

Differential privacy provides an intuitively understandable notion of privacy - a particular sample's inclusion or exclusion in the dataset does not change the probability of a particular outcome very much: it does so by a multiplicative factor of $e^\epsilon$ and an additive amount, $\delta$.

The following theorem, a proof of which can be found in [12], allows us to move the burden of differential privacy to the discriminator; the differential privacy of the generator will follow by the theorem.

**Theorem.** *(Post-processing) Let $\mathcal{M}$ be an $(\epsilon, \delta)$-differentially private algorithm and let $f : \mathcal{O} \to \mathcal{O}'$ where $\mathcal{O}'$ is any arbitrary space. Then $f \circ \mathcal{M}$ is $(\epsilon, \delta)$-differentially private.*

### 3.2 PRIVATE AGGREGATION OF TEACHER ENSEMBLES (PATE)

In this section we describe the PATE mechanism first defined in [25] and later improved upon by [26]. The PATE mechanism provides a differentially private method for classification, a core component of the GAN framework; the discriminator is a classifier trained to identify whether samples are real/fake.

In order to build a differentially private classifier, the dataset is first divided into $k$ *disjoint* subsets $\mathcal{D}_1, ..., \mathcal{D}_k$. $k$ classifiers, $T_1, ..., T_k$ (referred to as *teachers*) are then trained *separately* on the $k$ sub-datasets (i.e. $T_i$ is only trained on $\mathcal{D}_i$). Given a new input feature vector $\mathbf{x}$ to classify, the

differentially private output is given by passing $\mathbf{x}$ to each of the $k$ teachers, and then performing a *noisy* aggregation of the resulting outputs.

Formally, given the $k$ teachers, $m$ possible classes and an input feature vector, $\mathbf{x}$, set

$$n_j(\mathbf{x}) = |\{T_i : T_i(\mathbf{x}) = j\}| \text{ for } j = 1, ..., m \tag{3}$$

so that $n_j(\mathbf{x})$ is the number of teachers that output class $j$ for $\mathbf{x}$. The output of the PATE$_\lambda$ mechanism for input $\mathbf{x}$ is then defined as

$$\text{PATE}_\lambda(\mathbf{x}) = \underset{j \in [m]}{\arg\max}(n_j(\mathbf{x}) + Y_j) \tag{4}$$

where $Y_1, ..., Y_m$ are i.i.d. $Lap(\lambda)$ random variables. The following result, found in [25], follows from [12].

**Theorem.** *The output of a single query to the PATE$_\lambda$ mechanism is $(\frac{1}{\lambda}, 0)$-differentially private.*

In order to apply this framework in the GAN framework, however, we require that the discriminator be differentiable, which the *output* of this classification mechanism is not (note that accessing the internal parameters of the teachers would violate differential privacy, the only thing we have access to in this case is the output). Instead, we draw on the PATE extension (also introduced in [25]) in which a *student* model is trained. This student model (after being trained) is free to access, not only its outputs given inputs but also its internal parameters. The model itself is differentially private.

Formally, the student, $S$, is a classifier that is trained by taking some *public, unlabelled* data, $\mathcal{P} = \{\mathbf{x}_i\}_{i=1}^K$, passing each sample, $\mathbf{x}_i$, through the (standard) PATE mechanism, to receive a differentially private label, $\hat{y}_i$, and forming a new (noisy-)teacher-labelled dataset $\hat{\mathcal{P}} = \{(\mathbf{x}_i, \hat{y}_i)\}_{i=1}^K$ on which the student is then trained.

Importantly, we can make the student differentiable - it can be modelled using any classifier, such as a neural net. Moreover, querying the student is "free" - there is no privacy cost associated with passing an input to the student and receiving an output, the only privacy cost is in acquiring the data on which to train the student. We state the following result which follows from the analysis in [25].

**Theorem.** *The student, $S$, trained on the dataset $\hat{\mathcal{P}}$ where labels were generated according to the PATE$_\lambda$ mechanism using $\lambda = \frac{K}{2\epsilon}$, is $(\epsilon, 0)$-differentially private with respect to the original data $\mathcal{D}$.*

## 4 PROPOSED METHOD: PATE-GAN

The proposed method builds on GAN and PATE frameworks. We replace the GAN discriminator with a PATE mechanism so that our discriminator is differentially private, but require the (differentiable) student version to allow back-propagation to the generator. We modify the implementation of the student, noting that the training paradigm presented in [25] is not appropriate for this setting due to the lack of publicly available data. Before training, we partition the dataset into $k$ subsets, $\mathcal{D}_1, ..., \mathcal{D}_k$, with $|\mathcal{D}_i| = \frac{|\mathcal{D}|}{k}$ for $\forall i$.

### 4.1 GENERATOR

The generator, $G$, is as in the standard GAN framework. Formally it is a function $G(\cdot; \theta_G) : [0, 1]^d \to \mathcal{U}$, parametrized by $\theta_G$ that takes random noise, $\mathbf{z} \sim \text{Unif}([0, 1]^d)$, as input and outputs a vector in $\mathcal{U} = \mathcal{X} \times \mathcal{Y}$. The generator will be trained to minimize its loss with respect to the *student*-discriminator. Given $n$ i.i.d. samples of $\text{Unif}([0, 1]^d)$, $\mathbf{z}_1, ..., \mathbf{z}_n$, the empirical loss of $G$ at $\theta$ for fixed $S$ is defined by

$$\mathcal{L}_G(\theta_G; S) = \sum_{j=1}^n \log(1 - S(G(\mathbf{z}_j; \theta_G))). \tag{5}$$

We will denote by $\mathcal{P}_G$ the distribution induced by $G$ over $\mathcal{U}$.

### 4.2 DISCRIMINATOR

In the standard GAN framework, there is a single discriminator, $D$, that is trained in a directly adversarial fashion with $G$, where at each iteration either $G$ is trying to improve its loss with respect

to $D$ or $D$ is trying to improve its loss with respect to $G$. In our proposed model, however, we replace $D$ with the PATE mechanism. This means we introduce $k$ teacher-discriminators, $T^1, ..., T^k$, and a student discriminator, $S$. A noticeable difference is that the adversarial training is no longer symmetrical: the teachers are now being trained to improve their loss with respect to $G$ but $G$ is being trained to improve its loss with respect to the student $S$ which in turn is being trained to improve its loss with respect to the teachers.

### 4.2.1 TEACHER-DISCRIMINATORS

Formally, the teacher-discriminators (which we will refer to simply as teachers) are functions $T_1(\cdot; \theta_T^1), ..., T_k(\cdot; \theta_T^k) : \mathcal{U} \to [0, 1]$ each parametrized by $\theta_T^i$. The teachers are given either a real sample from their corresponding partition of the dataset (i.e. $T_i$ may receive a sample from $\mathcal{D}_i$) as input or a sample from the generator. The teachers are then trained to classify them.

Given $n$ i.i.d. samples of $\mathrm{Unif}([0, 1]^d)$, $\mathbf{z}_1, ..., \mathbf{z}_n$, we define the empirical loss of teacher $i$ with weights $\theta_T^i$ for fixed $G$ by

$$\mathcal{L}_T^i(\theta_T^i) = -\Big[ \sum_{\mathbf{u} \in \mathcal{D}_i} \log T_i(\mathbf{u}; \theta_T^i) + \sum_{j=1}^{n} \log(1 - T_i(G(\mathbf{z}_j); \theta_T^i)) \Big]. \tag{6}$$

Each teacher is trained in the same way the discriminator is trained in a standard GAN framework, except that here the teacher only ever sees its partition of the real data.

### 4.2.2 STUDENT-DISCRIMINATORS

The main innovation of our paper comes from our implementation of the student-discriminator (which we will refer to simply as the student) in this setting. The differential privacy guarantee provided by the standard student model is only with respect to the original data, $\mathcal{D}$, and not the public data, $\mathcal{P}$, used to train the student. In our setting, where the entire focus is on generating synthetic data *because no data is publicly available*, we must propose a novel methodology to train the student without public data.

We first note, that the student training paradigm described in [25] would involve training the student using data similar to that used to train the generator - i.e. by taking an equal number of samples from each and then labelling those using the standard $\mathrm{PATE}_\lambda$ mechanism (where here "labelling" refers to assigning them a real/fake label - not the label $y$ present in the data). We consider the implications of training the student on teacher-labelled generated samples only.

We first observe that during training of the generator, the discriminator is only evaluated on samples from the generator itself, and not the real data, so by training the student only on generated samples we are in fact training it on the distribution we need it to perform well on. However, we note that if the student only sees unrealistic samples from the generator (i.e. generated samples that most teachers label as fake), then the student will not contain any information that the generator can use to improve its generated samples. It is therefore important that some of the generated samples the student is trained on are realistic. We then note that if $\mathrm{Supp}(\mathcal{P}_U) \subset \mathrm{Supp}(\mathcal{P}_G)$ then some of the generated samples will be realistic.

In order to ensure $\mathrm{Supp}(\mathcal{P}_U) \subset \mathrm{Supp}(\mathcal{P}_G)$, we normalize the data into $[0, 1]^d$ and then initialize the parameters of the generator randomly using Xavier initialization. It follows that $\mathrm{Supp}(\mathcal{P}) \subset [0, 1]^d \subset G([0, 1]^d) = G(\mathrm{Supp}(\mathbf{Z})) = \mathrm{Supp}(G(\mathbf{Z}))$ when $\mathbf{Z} \sim \mathrm{Unif}([0, 1]^d)$.

We create our training data for the student by taking $n$ i.i.d. samples of $\mathrm{Unif}([0, 1]^d)$, $\mathbf{z}_1, ..., \mathbf{z}_n$, generating $n$ samples using the generator, $\hat{\mathbf{u}}_1, ..., \hat{\mathbf{u}}_n$ with $\hat{\mathbf{u}}_j = G(\mathbf{z}_j)$, and using the teachers to label these using $\mathrm{PATE}_\lambda$, setting $r_j = \mathrm{PATE}_\lambda(\hat{\mathbf{u}}_j)$. We train the student, $S(\cdot; \theta_S) : \mathcal{U} \to [0, 1]$, to maximize the standard cross-entropy loss on this teacher-labelled data, i.e.

$$\mathcal{L}_S(\theta_S) = \sum_{j=1}^{n} r_j \log S(\hat{\mathbf{u}}_j; \theta_S) + (1 - r_j) \log(1 - S(\hat{\mathbf{u}}_j; \theta_S)). \tag{7}$$

Although a priori the above mechanism does not appear to depend on the number of teachers, it should be noted that for fixed $\lambda$, more teachers results in the teacher-labelled dataset being less

noisy - the noise being added is smaller relative to the counts $n_j$. This introduces a trade-off - for a small number of teachers, the noise may be too large and thus render the output meaningless; with a larger number of teachers, less data can be used to train each teacher, which may also render the output meaningless, even though the noise has a smaller effect. Finding the right balance in this problem is key. In our experiments, we use $d$ real and $d$ generated samples to train each teacher where $d$ is the dimension of the input space. Although the utility of a single teacher may be low, by aggregating (even noisily) the resulting classifier actually has high utility. Moreover, by using a minimal number of samples for each teacher, the effect of any individual sample on the output is small (because there are more teachers and each sample can effect at most 1 teacher) which means that our differential privacy guarantees are tighter - if we used fewer teachers, the mechanism still assumes that, in the worst case, the presence (or absence) of a single sample can completely flip a teacher's vote and so we still need to add the same noise.

We train $G, T^1, ..., T^k$ and $S$ iteratively[1], with each iteration of $G$ consisting of first performing $n_T$ updates on all teachers, then performing $n_S$ updates of the student. We perform generator iterations until our privacy constraint, $\epsilon$, has been reached. A block diagram of PATE-GAN can be found in the Appendix.

To calculate the privacy of our algorithm we use the moments accountant method given in [25] to derive a data-dependent privacy guarantee at run-time. Details of its definition, and key results we use can be found in the Appendix. We denote the moments accountant of PATE-GAN by $\alpha(l)$. The moments accountant allows us to more tightly bound the total privacy cost of our mechanism than standard composition theorems would, and moreover attributes a lower privacy cost to accessing the noisy aggregation of the teachers when the teachers have a stronger consensus with the intuition being that when the teachers have a strong consensus, a single teacher (and therefore a single sample) has a much lower influence on the output than when the votes ($n_0$ and $n_1$) are close. Pseudo-code for PATE-GAN can be found in Algorithm 1.

We now state the main theorem of the paper, which follows from the theory in [25].

**Theorem 1.** *Algorithm 1, which takes as input $\delta > 0$, a dataset, $\mathcal{D}$, and outputs $G$ and $\epsilon$ is $(\epsilon, \delta)$-differentially private.*

The proof relies on applying the post-processing theorem where the discriminator corresponds to the mechanism $\mathcal{M}$ which takes outputs in $\mathcal{O}$ (in our case this corresponds to the weights of the discriminator), and the generator corresponds to the function $f$ which maps from $\mathcal{O}$ to $\mathcal{O}$ (which corresponds to the weights of the generator). For full details of the proof and further details of the theory required for it, see the Appendix.

## 5 EXPERIMENTS

In this section, we use a real-world Kaggle dataset (Credit card fraud detection dataset [11]) to evaluate PATE-GAN against the state-of-the-art benchmark (DPGAN [32]). In addition, we provide high-level (average) results for five additional datasets (with various characteristics): MAGGIC [27], UNOS-Heart wait-list [7], Kaggle cervical cancer dataset [16], UCI ISOLET dataset and UCI Epileptic Seizure Recognition dataset. A more detailed breakdown of the results for these datasets is given in the Appendix. Details of all six datasets can be also found in the Appendix.

### 5.1 EXPERIMENTAL SETTINGS

To empirically validate the quality of the generated dataset we introduce three different training-testing settings. *Setting A:* train the predictive models on the real training set, test the performance of the models on the real testing set. *Setting B:* train on the synthetic training set, test on the real testing set ([15]), *Setting C:* train on the synthetic training set, test on the synthetic testing set. Note that the training set and the testing set are disjoint in both the real and synthetic datasets.

We are interested in two comparisons. If we see a high predictive performance on the real data for models that were trained on synthetic data (Setting B), we can infer that the synthetic data has

---

[1]The teachers can be trained in parallel.

---

**Algorithm 1** Pseudo-code of PATE-GAN

---

1: **Input:** $\delta$, $\mathcal{D}$, $n_T$, $n_S$, batch size $n$, number of teachers $k$, noise size $\lambda$
2: **Initialize:** $\theta_G$, $\theta_T^1$, ..., $\theta_T^k$, $\theta_S$, $\alpha(l) = 0$ for $l = 1, ..., L$
3: Partition dataset into $k$ subsets $\mathcal{D}_1, ..., \mathcal{D}_k$ of size $\frac{|\mathcal{D}|}{k}$
4: **while** $\hat{\epsilon} < \epsilon$ **do**
5:     **for** $t_2 = 1, ..., n_T$ **do**
6:         Sample $\mathbf{z}_1, ..., \mathbf{z}_n \overset{\text{i.i.d.}}{\sim} \mathcal{P}_{\mathcal{Z}}$
7:         **for** $i = 1, ..., k$ **do**
8:             Sample $\mathbf{u}_1, ..., \mathbf{u}_n \overset{\text{i.i.d.}}{\sim} \mathcal{D}_i$
9:             Update teacher, $T_i$, using SGD
10:             $\nabla_{\theta_T^i} - \left[ \sum_{j=1}^d \log(T_i(\mathbf{u}_j)) + \log(1 - T_i(G(\mathbf{z}_j))) \right]$
11:     **for** $t_3 = 1, ..., n_S$ **do**
12:         Sample $\mathbf{z}_1, ..., \mathbf{z}_n \overset{\text{i.i.d.}}{\sim} \mathcal{P}_{\mathcal{Z}}$
13:         **for** $j = 1, ..., n$ **do**
14:             $\hat{\mathbf{u}}_j \leftarrow G(\mathbf{z}_j)$
15:             $r_j \leftarrow \text{PATE}_\lambda(\hat{\mathbf{u}}_i)$ for $j = 1, ..., n$
16:             Update moments accountant
17:             $q \leftarrow \frac{2 + \lambda |n_0 - n_1|}{4 \exp(\lambda |n_0 - n_1|)}$
18:             **for** $l = 1, ..., L$ **do**
19:                 $\alpha(l) \leftarrow \alpha(l) + \min\{2\lambda^2 l(l+1), \log((1-q)\left(\frac{1-q}{1-e^{2\lambda}q}\right)^l + qe^{2\lambda l})\}$
20:         Update the student, $S$, using SGD
21:         $\nabla_{\theta_S} - \sum_{j=1}^n r_j \log S(\hat{\mathbf{u}}_j) + (1 - r_j) \log(1 - S(\hat{\mathbf{u}}_j))$
22:     Sample $\mathbf{z}_1, ..., \mathbf{z}_n \overset{\text{i.i.d.}}{\sim} \mathcal{P}_{\mathcal{Z}}$
23:     Update the generator, $G$, using SGD
24:     $\nabla_{\theta_G} \left[ \sum_{i=1}^n \log(1 - S(G(\mathbf{z}_i))) \right]$
25:     $\hat{\epsilon} \leftarrow \min_l \frac{\alpha(l) + \log(\frac{1}{\delta})}{l}$
26: **Output:** $G$

---

captured the relationship between features and labels well. Moreover, synthetic data that does well in this setting can be used to train models without ever seeing the real data.

On the other hand, when we consider synthetic data for use in competitions such as Kaggle, we need synthetic data that allows researchers to do meaningful comparisons on the synthetic data. In this setting, the researchers will only be able to use the synthetic data as both the training and testing set, and will need to develop their algorithms using results on the synthetic data. Now it becomes important that the relative performance of two algorithms when trained and tested on the synthetic data (Setting C), is similar to their relative performance when trained and tested on the real data (Setting A). A simple requirement would be that if model 1 is better than model 2 on the real data, then model 1 is better than model 2 on the synthetic data. This allows researchers to use the synthetic data to choose the best method(s) to try on the real data (or rather to give to the data-holder to try on the real data).

For both comparisons, we use 12 different predictive models, shown in Table 1. We use two performance metrics to measure the capability of each model in predicting the label: (1) area under the receiver operating characteristics curve (AUROC), (2) area under the precision recall curve (AUPRC). Throughout the experiments we fix $\delta = 10^{-5}$ for use as input to PATE-GAN and DPGAN. We also report the performance of the original GAN framework ("GAN"), which serves to indicate an upper bound on performance and allows us to see how much performance is lost due to the two differential privacy mechanisms (PATE-GAN and DPGAN). The details of hyper-parameter optimization and benchmark implementations can be found in the Appendix.

| | AUROC | | | AUPRC | | |
|---|---|---|---|---|---|---|
| | GAN | **PATE-GAN** | DPGAN | GAN | **PATE-GAN** | DPGAN |
| Logistic Regression | 0.8950 | 0.8728 | 0.8720 | 0.4069 | 0.3907 | 0.3923 |
| Random Forests [5] | 0.9075 | 0.8980 | 0.8730 | 0.3219 | 0.3157 | 0.2926 |
| Gaussian Naive Bayes [29] | 0.8861 | 0.8817 | 0.8522 | 0.1963 | 0.1858 | 0.1601 |
| Bernoulli Naive Bayes [29] | 0.8997 | 0.8968 | 0.8891 | 0.2169 | 0.2099 | 0.2069 |
| Linear SVM [10] | 0.7611 | 0.7523 | 0.7502 | 0.4473 | 0.4466 | 0.4464 |
| Decision Tree [28] | 0.9102 | 0.9011 | 0.8647 | 0.4071 | 0.3978 | 0.3672 |
| LDA [3] | 0.8710 | 0.8510 | 0.8487 | 0.1956 | 0.1852 | 0.1788 |
| AdaBoost [17] | 0.9143 | 0.8952 | 0.8809 | 0.4530 | 0.4366 | 0.4234 |
| Bagging [4] | 0.8951 | 0.8877 | 0.8657 | 0.3303 | 0.3221 | 0.3073 |
| GBM [18] | 0.8848 | 0.8709 | 0.8499 | 0.3057 | 0.2974 | 0.2773 |
| Multi-layer Perceptron | 0.9086 | 0.8925 | 0.8787 | 0.4790 | 0.4693 | 0.4600 |
| XgBoost [8] | 0.9058 | 0.8904 | 0.8637 | 0.3837 | 0.3700 | 0.3440 |
| **Average** | **0.8866** | **0.8737** | **0.8578** | **0.3453** | **0.3351** | **0.3219** |

Table 1: Performance comparison of 12 different predictive models in Setting B (trained on synthetic, tested on real) in terms of AUROC and AUPRC (the generators of PATE-GAN and DPGAN are $(1, 10^{-5})$-differentially private).

## 5.2 RESULTS WITH SETTING B

In this subsection, we evaluate PATE-GAN and DPGAN in Setting B (trained on synthetic, tested on real) to understand whether or not the models are capturing the feature-label relationships well. Intuitively, if a synthetic dataset is such that a model trained on it performs well when performance is measured on real data, then the relationship between feature and label in the synthetic data is similar to that in the real data. In Table 1 we give the results for the Kaggle Credit dataset for all 12 predictive models. In Table 2, we give the performance on each dataset averaged across the 12 methods for each of the 6 datasets. A breakdown of the performance of each predictive model for each dataset can be found in the Appendix. Across all datasets, we see that PATE-GAN is capable of generating synthetic samples that better preserve the feature-label relationship (according to AUROC and AUPRC) than DPGAN.

| Datasets | AUROC | | | AUPRC | | |
|---|---|---|---|---|---|---|
| | GAN | PATE-GAN | DPGAN | GAN | PATE-GAN | DPGAN |
| Kaggle Credit | 0.8866 | 0.8737 | 0.8578 | 0.3453 | 0.3351 | 0.3219 |
| MAGGIC | 0.6574 | 0.6446 | 0.6286 | 0.3054 | 0.2952 | 0.2820 |
| UNOS | 0.6277 | 0.5996 | 0.5552 | 0.6554 | 0.6282 | 0.5862 |
| Kaggle Cervical Cancer | 0.9268 | 0.9108 | 0.8699 | 0.5994 | 0.5460 | 0.4851 |
| UCI ISOLET | 0.8171 | 0.6399 | 0.5577 | 0.5561 | 0.2953 | 0.2146 |
| UCI Epileptic Seizure Recognition | 0.9173 | 0.7681 | 0.6718 | 0.8133 | 0.6512 | 0.5369 |

Table 2: Performance comparison of 12 different predictive models in Setting B (trained on synthetic, tested on real) in terms of AUROC and AUPRC (the generators of PATE-GAN and DPGAN are $(1, 10^{-5})$-differentially private) over 6 different datasets. GAN is $(\infty, \infty)$-differentially private and is given to indicate an upper bound of PATE-GAN and DPGAN.

We note that the performance of all models, including the original GAN model (i.e. PATE-GAN - or equivalently DPGAN - with $(\infty, \infty)$-differential privacy) in the high dimensional UCI ISOLET and UCI Epileptic Seizure Recognition datasets is lower than in lower dimensional datasets (when

compared to the baseline AUROC and AUPRC found in the Appendix). We do, however, see that both PATE-GAN and DPGAN show more significant decreases in performance than the original GAN in these high-dimensional settings. In the case of PATE-GAN, we believe this may be due to the fact that the student discriminator is trained only using data from the generator, and therefore requires that some of the generated data look somewhat realistic from the start, which is a harder requirement to satisfy as the data has more dimensions. On the other hand, in DPGAN, noise must be added to each component of the gradient (of the discriminator) and so in higher dimensions the norm of the noise added is larger. Note that in PATE-GAN, noise is added only to the teacher outputs, whose dimension (typically 1) does not depend on the dimension of the input data, and so this phenomena does not present itself in PATE-GAN. The results on both the UCI datasets would suggest that the loss from increasing noise norm (for DPGAN) is greater than from difficulty in randomly generating realistic samples (for PATE-GAN).

### 5.3 VARYING THE PRIVACY CONSTRAINT ($\epsilon$)

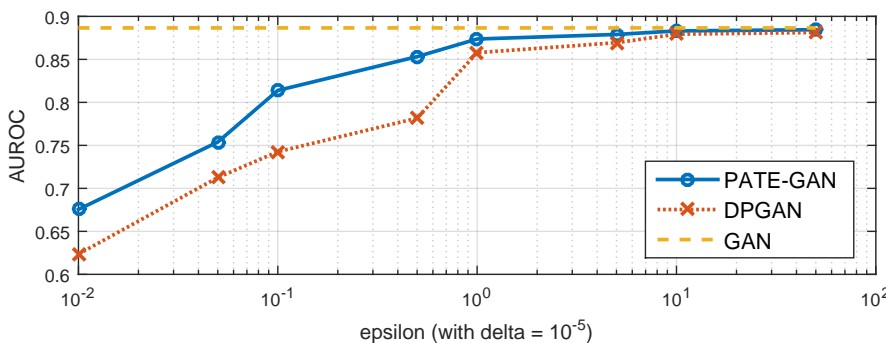

Figure 1: Average AUROC performance across 12 different predictive models trained on the synthetic dataset generated by PATE-GAN and DPGAN with various $\epsilon$ (with $\delta = 10^{-5}$) (Setting B).

In Fig. 1, we investigate the trade off between privacy constraint and utility. In the table we report the average performance of AUROC over the 12 different predictive models for PATE-GAN and the benchmark for various $\epsilon$ (with $\delta = 10^{-5}$). As can be seen in Fig. 1, PATE-GAN is consistently better than DPGAN over the entire range of tested $\epsilon$. We believe this is because the PATE mechanism allows us to more tightly bound the influence of a single sample on the discriminator, and hence we can provide tighter differential privacy guarantees - when the differential privacy guarantee is fixed, this results in higher quality synthetic data. Of course, as we increase $\epsilon$ (i.e. decrease the required privacy) both methods converge to the performance of GAN and the increase in performance of PATE-GAN over DPGAN becomes smaller.

### 5.4 SETTING A VS SETTING C: PRESERVING THE RANKING OF PREDICTIVE MODELS

As discussed at the beginning of this section, it is important that a synthetic dataset respects the ranking of models (in terms of their prediction performances) [21]. To evaluate this, we now introduce a new metric, which we refer to as the Synthetic Ranking Agreement (SRA). Suppose that we have $L$ predictive models, $f_1, f_2, ..., f_L$[2]. Furthermore, suppose that the performance of model $i$ when trained and tested on the real data (Setting A) is $A_i \in \mathbb{R}$ and that the performance of model $i$ when trained and tested on the synthetic data (Setting C) is $C_i \in \mathbb{R}$. Then we define the Synthetic Ranking Agreement by

$$\mathbf{SRA}(\{A_i\}_{i=1}^L, \{C_i\}_{i=1}^L) = \frac{1}{L(L-1)} \sum_{j=1}^L \sum_{k \neq j} \mathbb{I}\Big((A_j - A_k) \times (C_j - C_k) > 0\Big) \qquad (8)$$

where $\mathbb{I}$ is an indicator function. Note that the summand is 1 when the ordering of algorithms $j$ and $k$ are the same in both settings, and is 0 when the ordering in one setting differs from the ordering in the other.

---

[2]For the results in Table 3, we use the same 12 predictive models as used in Table 1

| | PATE-GAN | DPGAN | | | PATE-GAN | DPGAN |
|---|---|---|---|---|---|---|
| $\epsilon = 0.01$ | 0.6909 | 0.5273 | $\epsilon = 1$ | | 0.8364 | 0.8000 |
| $\epsilon = 0.05$ | 0.7455 | 0.6909 | $\epsilon = 5$ | | 0.8909 | 0.8364 |
| $\epsilon = 0.1$ | 0.7818 | 0.7455 | $\epsilon = 10$ | | 0.9091 | 0.8909 |
| $\epsilon = 0.5$ | 0.8000 | 0.7818 | $\epsilon = 50$ | | 0.9091 | 0.9091 |

Table 3: Synthetic Ranking Probability of PATE-GAN and the benchmark when comparing Setting A and Setting C for various $\epsilon$ (with $\delta = 10^{-5}$) in terms of AUROC. The Synthetic Ranking Agreement of Original GAN is 0.9091, which is also attained by both PATE-GAN and DPGAN for $\epsilon = 50$.

We compare the SRA of PATE-GAN and the benchmark for various $\epsilon$ (with $\delta = 10^{-5}$)[3]. As can be seen in Table 3, PATE-GAN achieves the best SRA across all values of $\epsilon$.

In the Appendix, we perform a similar experiment in which we compare the ranking of features by their importance (determined by their absolute Pearson correlation coefficient with the label) on the original dataset and on the synthetic dataset (generated by PATE-GAN and the benchmark) and report the results using a metric that is identical to SRA, with the model performances ($\{A_i\}, \{C_i\}$) substituted for feature importances.

## 5.5 QUANTITATIVE ANALYSIS ON THE NUMBER OF TEACHERS

The number of teachers is a hyper-parameter of PATE-GAN and we choose the number of teachers among $\{N/10, N/50, N/100, N/500, N/1000, N/5000, N/10000\}$ where $N$ is the total number of samples. As we described in the previous section, there is a trade-off between number of teachers and the corresponding quality of the synthetic data. Table 4 quantitatively shows the trade-off between the number of teachers and the performance (in terms of both AUROC and AUPRC).

| # of teachers | $N/10$ | $N/50$ | $N/100$ | $N/500$ | **N/1000** | $N/5000$ | $N/10000$ |
|---|---|---|---|---|---|---|---|
| AUROC | 0.5425 | 0.6398 | 0.7638 | 0.8343 | **0.8737** | 0.8655 | 0.8282 |
| AUPRC | 0.1273 | 0.2484 | 0.2900 | 0.3184 | **0.3351** | 0.3278 | 0.3092 |

Table 4: Trade-off between the number of teachers and the performances (AUROC, AUPRC)

## 6 DISCUSSION

In this paper we introduced a novel methodology for generating differentially private synthetic data. Through several experiments we demonstrated the ability of our method to produce high quality synthetic data while being able to give strict differential privacy guarantees.

In order to apply PATE to the GAN setting, we needed to use the original GAN framework. Extending PATE to the regression setting so that, for example, a Wasserstein GAN can be used instead, is an open and interesting question, and a potential direction for future research.

## ACKNOWLEDGEMENT

The authors would like to thank the reviewers for their helpful comments. The research presented in this paper was supported by the Office of Naval Research (ONR) and the NSF (Grant number: ECCS1462245, ECCS1533983, and ECCS1407712).

---

[3]The ordering of models according to Table 1 is in fact quite consistent - the average agreed ranking probability (now applied to different folds of the data, rather than real vs. synthetic data) is 0.9273 (for AUROC). The rankings used are therefore sufficiently stable for this to be a meaningful metric.

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

APPENDIX

THEORY REQUIRED FOR THEOREM 1

**Theorem 2.** *Algorithm 1, which takes as input $\delta > 0$, a dataset, $\mathcal{D}$, and outputs $G$ and $\epsilon$ is $(\epsilon, \delta)$-differentially private.*

In order to prove our theorem, we define the moments accountant [1] and state the theorems that the data-dependent privacy guarantees of the PATE mechanism rely on. For proofs of the results below, see [25] and the references therein.

**Definition 3.** *(Privacy Loss) Let $\mathcal{M}$ be a randomized algorithm taking outputs in a space $\mathcal{O}$ and $\mathcal{D}, \mathcal{D}'$ be neighbouring datasets. Let $aux$ denote an auxiliary input. For an outcome $o \in \mathcal{O}$, the privacy loss at $o$ is defined as:*

$$c(o; \mathcal{M}, aux, \mathcal{D}, \mathcal{D}') = \log \frac{\mathbb{P}(\mathcal{M}(aux, \mathcal{D}) = o)}{\mathbb{P}(\mathcal{M}(aux, \mathcal{D}') = o)}. \tag{9}$$

*The privacy loss random variable $C(\mathcal{M}, aux, \mathcal{D}, \mathcal{D}')$ is defined as $c(\mathcal{M}(\mathcal{D}); \mathcal{M}, aux, \mathcal{D}, \mathcal{D}')$, i.e. the random variable defined by evaluating the privacy loss at an outcome sampled from $\mathcal{M}(\mathcal{D})$.*

**Definition 4.** *(Moments accountant) Let $\mathcal{M}$ be a randomized algorithm. The moments accountant is defined as:*

$$\alpha_{\mathcal{M}}(l) = \max_{aux, \mathcal{D}, \mathcal{D}'} \alpha_{\mathcal{M}}(l; aux, \mathcal{D}, \mathcal{D}') \tag{10}$$

*where $\alpha_{\mathcal{M}}(l; aux, \mathcal{D}, \mathcal{D}') = \log \mathbb{E}(\exp(lC(\mathcal{M}, aux, \mathcal{D}, \mathcal{D}')))$ is the moment generating function of the privacy loss random variable and the $\max$ is taken over neighbouring datasets $\mathcal{D}, \mathcal{D}'$.*

**Theorem 3.** *(Composability) Suppose that an algorithm $\mathcal{M}$ consists of a sequence of adaptive algorithms $\mathcal{M}_1, ..., \mathcal{M}_k$ where $\mathcal{M}_i$ outputs in $\mathcal{O}_i$ and takes inputs from $\Pi_{j=1}^{i-1}\mathcal{O}_j$ as well as the dataset $\mathcal{D}$. Then for any output sequence $o_1, ..., o_{k-1}$ and any $l$*

$$\alpha_{\mathcal{M}}(l; \mathcal{D}, \mathcal{D}') = \sum_{i=1}^{k} \alpha_{\mathcal{M}_i}(l; o_1, ..., o_{i-1}, \mathcal{D}, \mathcal{D}') \tag{11}$$

*where $\alpha_{\mathcal{M}}$ is conditioned on each $\mathcal{M}_i$'s output being $o_i$.*

**Theorem 4.** *(Tail bound) Let $\mathcal{M}$ be a randomized algorithm. For any $\epsilon > 0$, $\mathcal{M}$ is $(\epsilon, \delta)$-differentially private for*

$$\delta = \min_l \exp(\alpha_{\mathcal{M}}(l) - l\epsilon). \tag{12}$$

The following theorem combines Theorems 2, 3 and Lemma 4 from [25].

**Theorem 5.** *(Data-dependent privacy guarantee for PATE) Let $\mathcal{M}$ be the PATE mechanism defined in Section 3 of the paper. Let $n_0, n_1$ be as defined in Equation 3 of the paper. Let $q = \frac{2 + \lambda|n_0 - n_1|}{4 \exp(\lambda|n_0 - n_1|)}$. Then*

$$\alpha_{\mathcal{M}}(l) \leq \min\{2\lambda^2 l(l+1), \log((1-q)\left(\frac{1-q}{1-e^{2\lambda}q}\right)^l + qe^{2\lambda l})\}. \tag{13}$$

*Proof of Theorem 2.* We use Theorem 5 to bound the moments accountant for each query to the PATE mechanism during the training of our algorithm (i.e. each time a generated sample is labeled by the teachers). Theorem 3 then allows us to sum the individual bounds for each query to bound the moments accountant of the entire algorithm. Theorem 4 then allows us to derive a value for $\epsilon$ given $\delta$. □

## DATA DESCRIPTIONS

### KAGGLE CREDIT CARD FRAUD DETECTION DATA DESCRIPTION

The Kaggle credit card fraud detection dataset [11] contains transactions made by credit cards in September 2013 by European cardholders and the label is whether or not the transaction is fraudulent. The total number of features is 29 (binary: 0, continuous: 29) and the number of samples in this dataset is 284,807. Among the 284,807 samples, 492 (0.2%) samples are fraudulent transactions.

### MAGGIC DATA DESCRIPTION

The Meta-analysis Global Group in Chronic Heart Failure (MAGGIC) dataset [27] is a collection of 30 different datasets from 30 different medical studies containing patients who experienced heart failure. We set the label of each patient as 1-year all-cause mortality, excluding all patients who are censored before 1-year. The total number of features is 29 (binary: 20, continuous: 9) and the number of patients in this dataset is 30,389. Among the 30,389 patients, 5,723 (18.8%) patients died within 1 year.

### UNOS DATA DESCRIPTION

The United Network for Organ Transplantation (UNOS) dataset [7] provides information about all patients in the U.S. who have received a transplantation or were on the wait-list during the period 1985-2015. In this paper, we focus on the patients who were on the heart transplant wait-list. The objective is to predict 1-year all-cause mortality. The total number of features is 20 (binary: 18, continuous: 2) and the number of patients in this dataset is 23,706. Among the 23,706 patients, 12,606 (53.2%) patients died within 1 year.

### KAGGLE CERVICAL CANCER DATA DESCRIPTION

The Kaggle cervical cancer dataset [16] was collected at 'Hospital Universitario de Caracas' in Caracas, Venezuela. It contains demographic information, habits, and historic medical records. The total number of features is 35 (binary: 24, continuous: 11) and the number of patients in this dataset is 858. Among the 858 patients, 55 (6.4%) patients have positive biopsy.

### UCI ISOLET DATA DESCRIPTION

The UCI ISOLET dataset `https://archive.ics.uci.edu/ml/datasets/isolet` was generated by speaking the name of each letter of the alphabet. The task is to classify each spoken letter as either a vowel or a consonant (binary classification). The total number of features is 617 and the number of samples in this dataset is 7797. Among the 7797 samples, 1500 (19.2%) samples are vowels.

### UCI EPILEPTIC SEIZURE RECOGNITION DATA DESCRIPTION

The UCI Epileptic Seizure Recognition dataset `https://archive.ics.uci.edu/ml/datasets/Epileptic+Seizure+Recognition` was generated by recording brain activity. The task is to classify activity as seizure activity (binary classification). The total number of features is 179 and the number of samples in this dataset is 11500. Among the 11500 samples, 2300 (20.0%) samples correspond to seizure activity.

DATA SUMMARY AND SETTING A PERFORMANCE

Table 5 summarises the 6 datasets we use and provides a baseline performance for a predictive model on each dataset - recall that Setting A refers to training and testing on the real data. The AUROC and AUPRC in this setting are upper bounds on the AUROC and AUPRC we could hope to achieve in Setting B.

| Datasets | No of samples | No of features | AUROC | AUPRC |
|---|---|---|---|---|
| Kaggle Credit | 284807 | 29 | 0.9438 | 0.7020 |
| MAGGIC | 30389 | 29 | 0.7069 | 0.3638 |
| UNOS | 23706 | 20 | 0.6416 | 0.6677 |
| Kaggle Cervical cancer | 858 | 35 | 0.9354 | 0.6314 |
| UCI ISOLET | 7797 | 617 | 0.9671 | 0.8758 |
| UCI Epileptic Seizure Recognition | 11500 | 179 | 0.9809 | 0.9511 |

Table 5: No of samples, No of features, Average AUROC and AUPRC performance across 12 different predictive models trained and tested on the real data (Setting A) for the 6 datasets: Kaggle Credit, MAGGIC, UNOS, Kaggle Cervical Cancer, UCI ISOLET, UCI Epileptic Seizure Recognition.

HYPER-PARAMETER OPTIMIZATION

In all experiments, the depth of the generator and discriminator (student-discriminator in our case) in both PATE-GAN and the DPGAN benchmark [32] is set to 3. The depth of the teacher discriminators is set to 1. The number of hidden nodes in each layer is $d$, $d/2$ and $d$ (where $d$ is the feature dimension), respectively. We use *relu* as the activation functions of each layer except for the output layer where we use the *sigmoid* activation function and the batch size is 64 for both the generator and discriminator. We set $n_T = n_S = 5$. Using cross validation, we select the number of teachers, $k$, among N/10 N/50 N/100 N/500 N/1000 N/5000 N/10000. The learning rate is $10^{-4}$ and we use Adam Optimizer to minimize the loss function.

We use tensorflow to implement PATE-GAN and DPGAN. For DPGAN we use the code from the following link: `https://github.com/illidanlab`. We use the sklearn package in python to implement the 12 predictive models: Logistic Regression (*LogisticRegression*), Random Forests (*RandomForestClassifier*), Gaussian Naive Bayes (*GaussianNB*), Bernoulli Naive Bayes (*BernoulliNB*), Linear Support Vector Machine (*svm*), Decision Tree (*DecisionTree*), Linear Discriminant Analysis Classifier (*LinearDiscriminantAnalysis*), Adaptive Boosting (AdaBoost) (*AdaBoostClassifier*), Bootstrap Aggregating (Bagging) (*BaggingClassifier*), Gradient Boosting Machine (GBM) (*GradientBoostingClassifier*), Multi-layer Perceptron (*MLPClassifier*), and XgBoost (*XGBoostRegressor*).

ADDITIONAL RESULTS

VARYING THE PRIVACY CONSTRAINT ($\epsilon$) IN TERMS OF AUPRC

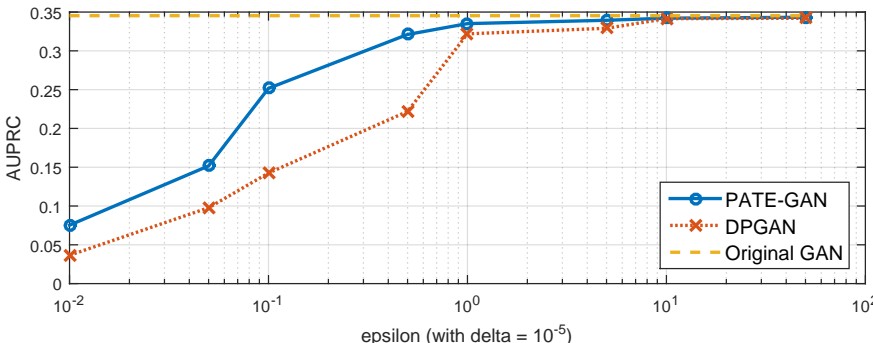

Figure 2: Average AUPRC performance across 12 different predictive models trained on the synthetic dataset generated by PATE-GAN and DPGAN with various $\epsilon$ (with $\delta = 10^{-5}$) (Setting B).

Similar to Fig. 1 in the main manuscript, Fig. 2 shows the trade off between the privacy constraint and utility, where utility is now measured in terms of AUPRC (rather than AUROC). We report the average performance in terms of AUPRC over the 12 different predictive models for PATE-GAN and the benchmark for various $\epsilon$ (with $\delta = 10^{-5}$). As can be seen in Fig. 2, PATE-GAN consistently outperforms DPGAN over the entire range of tested $\epsilon$ in terms of AUPRC as well.

PRESERVING THE RANKING OF VARIABLE IMPORTANCE IN KAGGLE CREDIT DATASET

We compare the ranking of variables by their importance (according to absolute Pearson correlation coefficient with the label) on the original dataset and on both synthetic datasets. We report the results using agreed ranking probability. As can be seen in Table 6, PATE-GAN achieves consistently better agreed ranking probability across all values of tested $\epsilon$ (with $\delta = 10^{-5}$).

|  | PATE-GAN | DPGAN |  | PATE-GAN | DPGAN |
|---|---|---|---|---|---|
| $\epsilon = 0.01$ | 0.8810 | 0.7963 | $\epsilon = 1$ | 0.9048 | 0.8783 |
| $\epsilon = 0.05$ | 0.8968 | 0.8148 | $\epsilon = 5$ | 0.9127 | 0.8915 |
| $\epsilon = 0.1$ | 0.8968 | 0.8333 | $\epsilon = 10$ | 0.9153 | 0.8942 |
| $\epsilon = 0.5$ | 0.9021 | 0.8545 | $\epsilon = 50$ | 0.9153 | 0.9021 |

Table 6: Agreed ranking probability of PATE-GAN and the benchmark to order the features by variable importance in terms of absolute Pearson correlation coefficient

HIGH-DIMENSIONAL RESULTS: UCI ISOLET AND UCI EPILEPTIC SEIZURE RECOGNITION

| $(\epsilon, \delta)$ | AUROC | | | AUPRC | | |
|---|---|---|---|---|---|---|
| | GAN | PATE-GAN | DPGAN | GAN | PATE-GAN | DPGAN |
| $(10, 10^{-5})$ | 0.8171 | 0.7688 | 0.7390 | 0.5561 | 0.4734 | 0.3831 |
| $(1, 10^{-5})$ | | 0.6399 | 0.5577 | | 0.2953 | 0.2146 |

Table 7: Average AUROC and AUPRC performance of 12 different predictive models trained on the synthetic datasets for $\epsilon = 1, 10$ with $\delta = 10^{-5}$ - Setting B using UCI ISOLET dataset. GAN is $(\infty, \infty)$-differentially private and is given to indicate an upper bound of PATE-GAN and DPGAN.

| $(\epsilon, \delta)$ | AUROC | | | AUPRC | | |
|---|---|---|---|---|---|---|
| | GAN | PATE-GAN | DPGAN | GAN | PATE-GAN | DPGAN |
| $(10, 10^{-5})$ | 0.9173 | 0.8718 | 0.8189 | 0.8133 | 0.7662 | 0.7201 |
| $(1, 10^{-5})$ | | 0.7681 | 0.6718 | | 0.6512 | 0.5369 |

Table 8: Average AUROC and AUPRC performance across 12 different predictive models trained on the synthetic datasets with various $\epsilon = 1, 10$ with $\delta = 10^{-5}$ - Setting B using UCI Epileptic Seizure Recognition dataset. GAN is $(\infty, \infty)$-differentially private and is given to indicate an upper bound of PATE-GAN and DPGAN.

MAGGIC DATASET RESULT

| | AUROC | | | AUPRC | | |
|---|---|---|---|---|---|---|
| | GAN | **PATE-GAN** | DPGAN | GAN | **PATE-GAN** | DPGAN |
| Logistic Regression | 0.6645 | 0.6413 | 0.6415 | 0.3113 | 0.2951 | 0.2967 |
| Random Forests | 0.6492 | 0.6397 | 0.6147 | 0.2953 | 0.2891 | 0.2660 |
| Gaussian Naive Bayes | 0.6770 | 0.6726 | 0.6431 | 0.3258 | 0.3153 | 0.2896 |
| Bernoulli Naive Bayes | 0.6647 | 0.6618 | 0.6541 | 0.3008 | 0.2938 | 0.2908 |
| Linear SVM | 0.6410 | 0.6301 | 0.6322 | 0.2911 | 0.2904 | 0.2902 |
| Decision Tree | 0.6689 | 0.6598 | 0.6234 | 0.3163 | 0.3070 | 0.2764 |
| LDA | 0.6656 | 0.6433 | 0.6456 | 0.3118 | 0.2950 | 0.3014 |
| AdaBoost | 0.6524 | 0.6333 | 0.6190 | 0.3054 | 0.2890 | 0.2758 |
| Bagging | 0.6454 | 0.6380 | 0.6160 | 0.2912 | 0.2830 | 0.2682 |
| GBM | 0.6609 | 0.6470 | 0.6260 | 0.3106 | 0.3023 | 0.2822 |
| Multi-layer Perceptron | 0.6390 | 0.6229 | 0.6091 | 0.2921 | 0.2824 | 0.2731 |
| XgBoost | 0.6604 | 0.6450 | 0.6183 | 0.3133 | 0.2996 | 0.2736 |
| **Average** | **0.6574** | **0.6446** | **0.6286** | **0.3054** | **0.2952** | **0.2820** |

Table 9: Performance comparison of 12 different predictive models in Setting B (trained on synthetic, tested on real) in terms of AUROC and AUPRC (the generators of PATE-GAN and DPGAN are $(1, 10^{-5})$-differentially private). GAN is $(\infty, \infty)$-differentially private and is given to indicate an upper bound of PATE-GAN and DPGAN.

UNOS HEART WAIT DATASET RESULT

| | AUROC | | | AUPRC | | |
|---|---|---|---|---|---|---|
| | GAN | **PATE-GAN** | DPGAN | GAN | **PATE-GAN** | DPGAN |
| Logistic Regression | 0.6407 | 0.6155 | 0.5548 | 0.6691 | 0.6450 | 0.5901 |
| Random Forests | 0.6159 | 0.5950 | 0.5574 | 0.6436 | 0.6129 | 0.5768 |
| Gaussian Naive Bayes | 0.6323 | 0.6015 | 0.5343 | 0.6648 | 0.6371 | 0.5824 |
| Bernoulli Naive Bayes | 0.6213 | 0.6045 | 0.5763 | 0.6501 | 0.6363 | 0.6077 |
| Linear SVM | 0.6244 | 0.5979 | 0.5581 | 0.6486 | 0.6254 | 0.5892 |
| Decision Tree | 0.6209 | 0.6019 | 0.5590 | 0.6496 | 0.6284 | 0.5819 |
| LDA | 0.6403 | 0.6077 | 0.5530 | 0.6682 | 0.6406 | 0.5882 |
| AdaBoost | 0.6222 | 0.5928 | 0.5527 | 0.6404 | 0.6194 | 0.5826 |
| Bagging | 0.6084 | 0.5858 | 0.5493 | 0.6325 | 0.6074 | 0.5693 |
| GBM | 0.6374 | 0.6040 | 0.5585 | 0.6679 | 0.6352 | 0.5920 |
| Multi-layer Perceptron | 0.6328 | 0.5927 | 0.5562 | 0.6629 | 0.6240 | 0.5856 |
| XgBoost | 0.6362 | 0.5956 | 0.5533 | 0.6676 | 0.6267 | 0.5880 |
| **Average** | **0.6277** | **0.5996** | **0.5552** | **0.6554** | **0.6282** | **0.5862** |

Table 10: Performance comparison of 12 different predictive models in Setting B (trained on synthetic, tested on real) in terms of AUROC and AUPRC (the generators of PATE-GAN and DPGAN are $(1, 10^{-5})$-differentially private). GAN is $(\infty, \infty)$-differentially private and is given to indicate an upper bound of PATE-GAN and DPGAN.

KAGGLE CERVICAL CANCER DATASET RESULT

| | AUROC | | | AUPRC | | |
|---|---|---|---|---|---|---|
| | GAN | **PATE-GAN** | DPGAN | GAN | **PATE-GAN** | DPGAN |
| Logistic Regression | 0.9188 | 0.9102 | 0.8945 | 0.5949 | 0.5605 | 0.4672 |
| Random Forests | 0.9515 | 0.9373 | 0.9237 | 0.6366 | 0.6361 | 0.5735 |
| Gaussian Naive Bayes | 0.9393 | 0.8890 | 0.7973 | 0.5605 | 0.4422 | 0.3702 |
| Bernoulli Naive Bayes | 0.8421 | 0.8331 | 0.8296 | 0.2491 | 0.2211 | 0.2160 |
| Linear SVM | 0.9282 | 0.9086 | 0.9050 | 0.6031 | 0.5921 | 0.5665 |
| Decision Tree | 0.9451 | 0.9434 | 0.9283 | 0.6455 | 0.6094 | 0.5734 |
| LDA | 0.9358 | 0.9155 | 0.8667 | 0.6518 | 0.6061 | 0.5629 |
| AdaBoost | 0.9361 | 0.8898 | 0.7989 | 0.6881 | 0.5587 | 0.4281 |
| Bagging | 0.9425 | 0.9275 | 0.9080 | 0.6257 | 0.5871 | 0.5809 |
| GBM | 0.9398 | 0.9333 | 0.9017 | 0.6927 | 0.6165 | 0.5422 |
| Multi-layer Perceptron | 0.9005 | 0.9064 | 0.7933 | 0.5675 | 0.5246 | 0.3746 |
| XgBoost | 0.9408 | 0.9351 | 0.8919 | 0.6784 | 0.5978 | 0.5657 |
| **Average** | **0.9268** | **0.9108** | **0.8699** | **0.5994** | **0.5460** | **0.4851** |

Table 11: Performance comparison of 12 different predictive models in Setting B (trained on synthetic, tested on real) in terms of AUROC and AUPRC (the generators of PATE-GAN and DPGAN are $(1, 10^{-5})$-differentially private). GAN is $(\infty, \infty)$-differentially private and is given to indicate an upper bound of PATE-GAN and DPGAN.

BLOCK DIAGRAMS

PATE-GAN

The two figures below indicate the iterative training procedure carried out by PATE-GAN; the figures correspond to a single generator update.

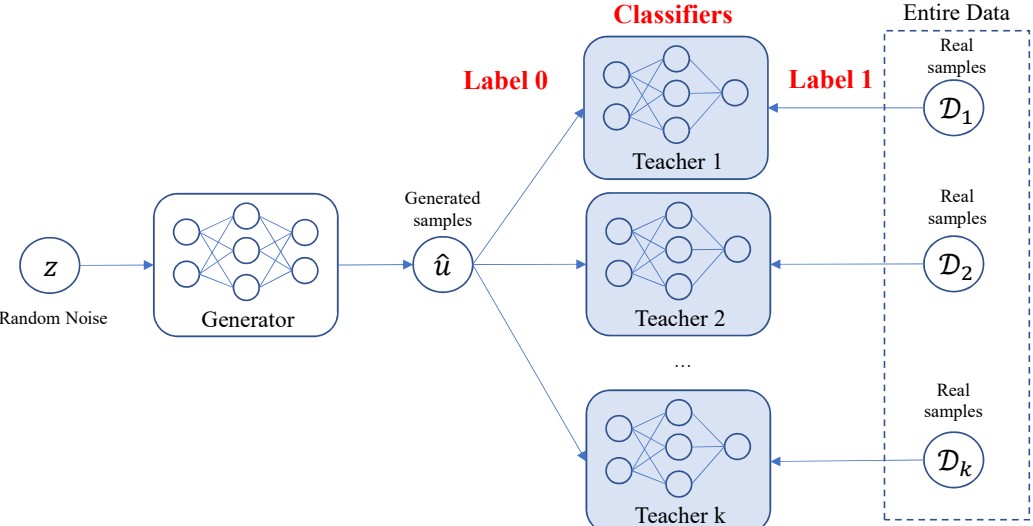

Figure 3: Block diagram of the training procedure for the teacher-discriminator during a single generator iteration. Teacher-discriminators are trained to minimize the classification loss when classifying samples as real samples or generated samples. During this step only the parameters of the teachers are updates (and not the generator).

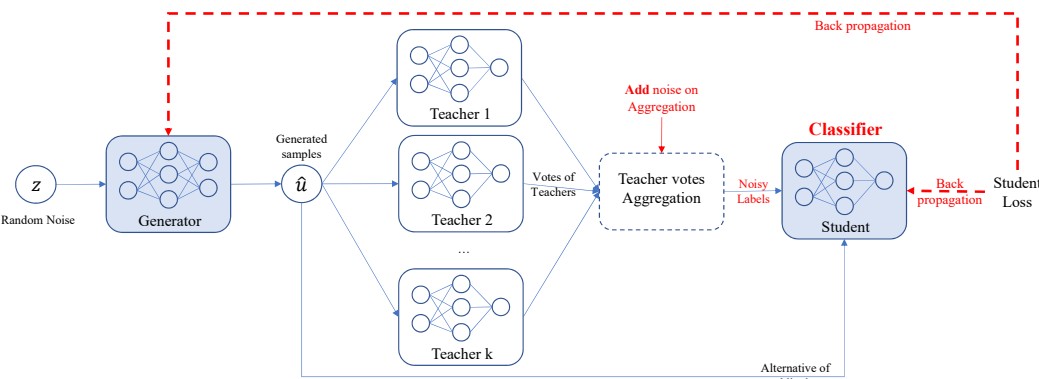

Figure 4: Block diagram of the training procedure for the student-discriminator and the generator. The student-discriminator is trained using noisy teacher-labelled generated samples (the noise provides the DP guarantees). The student is trained to minimize classification loss on this noisily labelled dataset, while the generator is trained to maximize the student loss. Note that the teachers are not updated during this step, only the student and the generator.

DPGAN [32]

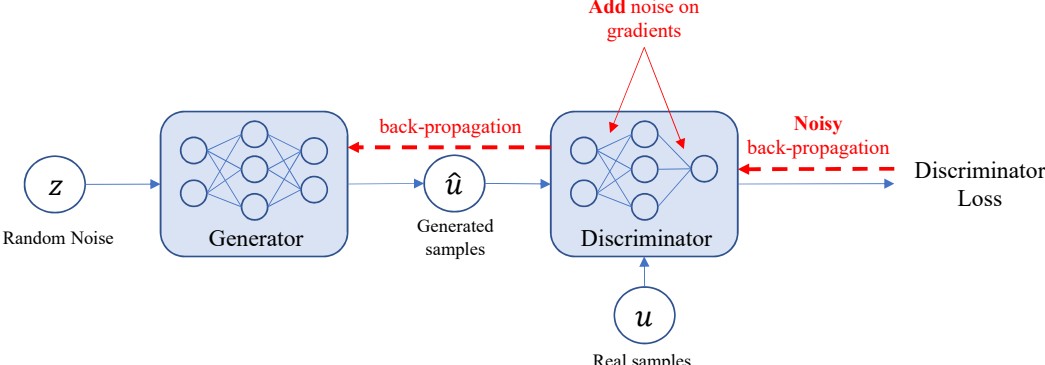

Figure 5: Block diagram of the DPGAN benchmark. It uses the standard WGAN framework. To guarantee differential privacy of the generator (with Post-processing Theorem), noise is added to the gradient of the discriminator during training to create a differentially private discriminator.

