# OpenReview forum: "PATE-GAN: Generating Synthetic Data with Differential Privacy Guarantees"
_ICLR.cc/2019/Conference_

### Official Review · AnonReviewer1 · 2018-11-03
**Differenntially  private synthetic data set generation via combining the PATE framework and GAN**

**Rating:** 7
**Confidence:** 3

**Review:**

The paper studies the problem of generating synthetic datasets (while ensuring differential privacy) via training a GAN. One natural approach is the teacher-student framework considered in the PATE framework.  In the original PATE framework, while the teachers are ensured to preserve differential privacy, the student model (typically a GAN) requires the presence of publicly data samples. The main contribution of this paper is to get around the requirement of public data via using uniformly random samples in [0,1]^d.

Differentially private synthetic data generation is clearly an important and a long-standing open problem. Recently, there has been some work on exploiting differentially private variants of GANs to generate synthetic data. However, the scale of these results is far from satisfactory. The current paper claims to bypass this issue by using the PATE-GAN approach.

I am not an expert on deep learning. The idea of bypassing the use of public data by taking uniformly random samples seems interesting. In my view, these random vectors are used in the GAN as some sort of a basis. It is interesting to see if this result extends to high-dimensional settings (i.e., where d  is very large).

---

> ### Author Response · Authors · 2018-11-14
> **RE: Differenntially private synthetic data set generation via combining the PATE framework and GAN**
>
> Thank you for your insightful comments.
>
> A1: We have performed simulations using higher-dimensional data which we will include in the paper. Specifically, we used two UCI datasets (UCI ISOLET dataset (dimensions: 617, no of samples: 7797, task: classify consonant vs vowel) and UCI Epileptic Seizure Recognition dataset (dimensions: 179, no of samples: 11500, task: classify seizure activity)) and varied the dimensionality to demonstrate the scalability of our method. The results on the full dataset (all 617 features and 179 features) can be seen in the following tables:
>
> (1) UCI ISOLET Dataset
> --------------------------------------------------------------------------------------------------
> (epsilon, delta) = (10, 10^-5) |      GAN     |     PATE-GAN     |     DPGAN    |
> --------------------------------------------------------------------------------------------------
>                AUROC                       |     0.817     |        0.769          |       0.739      |
>                AUPRC                       |     0.556     |        0.473          |       0.383      |
> --------------------------------------------------------------------------------------------------
>
> (2) UCI Epileptic Seizure Recognition Dataset
> --------------------------------------------------------------------------------------------------
> (epsilon, delta) = (10, 10^-5) |      GAN     |     PATE-GAN     |     DPGAN    |
> --------------------------------------------------------------------------------------------------
>                AUROC                       |     0.917     |        0.872          |       0.819      |
>                AUPRC                       |     0.813      |        0.766          |       0.720      |
> --------------------------------------------------------------------------------------------------
>
> As can be seen in the table, PATE-GAN works well in high-dimensional data continuing to outperform DPGAN. Detailed results with various dimensionalities will be added to the revised manuscript.

---

### Official Review · AnonReviewer2 · 2018-11-05
**Need improvement**

**Rating:** 6
**Confidence:** 4

**Review:**

[Post revision update] The authors' comments addressed my concerns, especially on the experiment side. I changed the score.

This paper applies the PATE framework to GAN, and evaluates the quality of the generated data with some predictive tasks. The experimental results on some real datasets show that the proposed algorithm outperforms DPGAN, and the generated synthetic data is quite useful in comparison with real data.
The presentation is clear and easy to follow. However, I think the paper needs to be improved in its novelty, and the techniques and experiments need to be more thorough.

More details:
- It might be necessary to consider using Gaussian noise[24] in replace of the Laplace noise, which, according to [24], would improve privacy and accuracy.
- This paper:
“Privacy-preserving generative deep neural networks support clinical data sharing” by Brett K. Beaulieu-Jones, Zhiwei Steven Wu, Chris Williams, Casey S. Greene
seems quite relevant. If so, you may want to add some discussion in the related work section or compare with their result.
- The last paragraph of the related works section mentioned some related work with shortcomings as working only on low-dimensional data and features of specific types, yet the experiments are also mostly done on low-dimensional datasets. I think it would be better to do a thorough evaluation on data of different kinds, such as image data.
- If the two evaluation metrics for private GAN is considered an important contribution of the paper, it might be better to make it a separate section and elaborate more on the motivation and method.
- It might be better to move some details (for example, instead of presenting the results of the 12 predictive models, presenting only the average, as it’s not very important how each of them performs) of the credit card fraud detection dataset to the appendix and bring the results of the other datasets to the main body.

---

> ### Author Response · Authors · 2018-11-14
> **RE: Need improvement**
>
> Thank you for your insightful comments.
>
> A1: Our key contribution is in building on PATE, and developing a new framework which can be used in the GAN setting. While using Gaussian noise may indeed improve our results further, the additional analysis required to use Gaussian noise for PATE is more involved (as noted in [24]), and its inclusion may therefore distract the reader from the main contribution of our paper.
>
> A2: The method proposed in this paper is very similar to (if not the same as) DPGAN. We will modify the related works section to read:
> “… The key idea is that noise is added to the gradient of the discriminator during training to create differential privacy guarantees. These ideas are also used in [Privacy-preserving generative deep neural networks support clinical data sharing]. Our method…”
>
> A3: A key difference between our work and these works, which we will highlight in the paper, is that they do not use differential privacy. In addition, we have performed simulations using higher-dimensional data which we will include in the paper. Specifically, we used two UCI datasets (UCI ISOLET dataset (dimensions: 617, no of samples: 7797, task: classify consonant vs vowel) and UCI Epileptic Seizure Recognition dataset (dimensions: 179, no of samples: 11500, task: classify seizure activity)) and varied the dimensionality to demonstrate the scalability of our method. The results on the full dataset (all 617 features and 179 features) can be seen in the following tables.
>
> (1) UCI ISOLET Dataset
> --------------------------------------------------------------------------------------------------
> (epsilon, delta) = (10, 10^-5) |      GAN     |     PATE-GAN     |     DPGAN    |
> --------------------------------------------------------------------------------------------------
>                AUROC                       |     0.817     |        0.769          |       0.739      |
>                AUPRC                       |     0.556     |        0.473          |       0.383      |
> --------------------------------------------------------------------------------------------------
>
> (2) UCI Epileptic Seizure Recognition Dataset
> --------------------------------------------------------------------------------------------------
> (epsilon, delta) = (10, 10^-5) |      GAN     |     PATE-GAN     |     DPGAN    |
> --------------------------------------------------------------------------------------------------
>                AUROC                       |     0.917     |        0.872          |       0.819      |
>                AUPRC                       |     0.813      |        0.766          |       0.720      |
> --------------------------------------------------------------------------------------------------
>
> As can be seen in the tables, PATE-GAN works well also in high-dimensional data and continues to outperform DPGAN. Detailed results will be added to the revised manuscript.
>
> A4: Thank you for the suggestion, the new metric that we are proposing is the agreed ranking probability of section 5.4. To highlight this we will move its introduction to the end of section 4, highlighting that it is one of our contributions.
>
> A5: Thank you for this suggestion, we will move the average results for the other datasets into the main manuscript. We will keep the 12 predictive models in the main manuscript for the Kaggle credit card fraud dataset, as we feel it gives a more complete picture of our results.

---

### Official Review · AnonReviewer3 · 2018-11-06
**Interesting setup, surprising that it works**

**Rating:** 7
**Confidence:** 4

**Review:**

This paper considers using a GAN to generate synthetic data in a differentially private manner [see also https://www.biorxiv.org/content/early/2018/06/05/159756 ]. The key novelty is the integration of the PATE differential privacy framework from recent work. Specifically, rather than a single distinguisher as is usual in a GAN, there is a "student distinguisher" and several "teacher distinguishers". The student distinguisher is used as usual except that it does not have access to the real data, only the teacher distinguishers have access to the real data (as well as the synthetic data). The data is partitioned amongst the teacher distinguishers and their output is aggregated in a differentially private manner (and gradients are not revealed). The role of the teacher distinguishers is solely to correct the student distinguisher when it errs.

What is strange about this setup is that the generator's only feedback is from the gradients of the student distinguisher, which is never exposed to the real data. The entire training process relies on the generator producing realistic data by chance at which point the teacher distinguishers can provide positive feedback. (The paper remarks about this in the middle of page 5.) It's surprising that this works, but there are experimental results to back it up.

I think it would be appropriate to remark that generating private synthetic data is known to be hard in the worst case [ https://eccc.weizmann.ac.il/report/2010/017/ ] and therefore it is necessary to use techniques like GANs.

Overall, I think the paper is interesting, well written, novel, and therefore appropriate for ICLR.

---

> ### Author Response · Authors · 2018-11-14
> **RE: Add Interesting setup, surprising that it works**
>
> Thank you for your insightful comments.
>
> A1: We note that the generator need only generate samples that are “somewhat” more realistic than other samples, thus providing the discriminator with some side information about what direction to guide the generator in. We also considered starting the student training using uniformly drawn samples from [0,1]^d and then transitioning to generator-only samples after the generator had a chance to start generating realistic samples but found this to be unnecessary. To further demonstrate this, we have now included results for higher-dimensional data in which it would be harder for the generator to do this “by chance”, and continue to show high performance.
>
> Specifically, we used two UCI datasets (UCI ISOLET dataset (dimensions: 617, no of samples: 7797, task: classify consonant vs vowel) and UCI Epileptic Seizure Recognition dataset (dimensions: 179, no of samples: 11500, task: classify seizure activity)) and varied the dimensionality to demonstrate the scalability of our method. The results on the full dataset (all 617 features and 179 features) can be seen in the following tables.
>
> (1) UCI ISOLET Dataset
> --------------------------------------------------------------------------------------------------
> (epsilon, delta) = (10, 10^-5) |      GAN     |     PATE-GAN     |     DPGAN    |
> --------------------------------------------------------------------------------------------------
>                AUROC                       |     0.817     |        0.769          |       0.739      |
>                AUPRC                       |     0.556     |        0.473          |       0.383      |
> --------------------------------------------------------------------------------------------------
>
> (2) UCI Epileptic Seizure Recognition Dataset
> --------------------------------------------------------------------------------------------------
> (epsilon, delta) = (10, 10^-5) |      GAN     |     PATE-GAN     |     DPGAN    |
> --------------------------------------------------------------------------------------------------
>                AUROC                       |     0.917     |        0.872          |       0.819      |
>                AUPRC                       |     0.813      |        0.766          |       0.720      |
> --------------------------------------------------------------------------------------------------
>
> As can be seen in the tables, PATE-GAN works well in high-dimensional data continuing to outperform DPGAN. Detailed results will be added to the revised manuscript.
>
> A2: We will add the following line to the end of the related works section:
> “Finally, it is worth remarking that it is known to be hard in the worst-case to generate private synthetic data [the above-mentioned paper] and techniques such as GANs are necessary to address this challenge.

---

### Public Comment · ~Chun-Hao_Chang1 · 2018-11-08
**Missing setting A performance**

Thank you for this great work!

I am wondering what's the performance of setting A (train in real training set, evaluate in real test set) across all 4 experiments. Paper only shows the ranking compared to setting C. It definitely helps the reader to evaluate if this privacy GAN actually capture the real characteristics of the original dataset, especially in 4 different experients with varying size of dataset. I will expect the gap between setting A and C should go larger as the dataset gets smaller.

---

> ### Author Response · Authors · 2018-11-14
> **RE: Missing setting A performance**
>
> Thank you for the comments.
>
> Please see the table below, containing the performance of setting A across all 4 experiments (average performance across all 12 predictive models). You can compare the below with Table 1, 5, 6, and 7 in Setting B.
> -------------------------------------------------------------------------------------
> Setting A Performance | No of samples  |  AUROC  |  AUPRC |
> -------------------------------------------------------------------------------------
>         Kaggle Credit          |        284,807       |  0.9438   |  0.7020  |
>            MAGGIC                |        30,389         |  0.7069   |  0.3638  |
>              UNOS                  |        23,706          |  0.6416   |  0.6677  |
>         Kaggle Cervical       |         858              |  0.9354   |  0.6314  |
> ------------------------------------------------------------------------------------

---

> > ### Public Comment · ~Chun-Hao_Chang1 · 2018-11-18
> > **Surprising differences in Kaggle Credit**
> >
> > Surprisingly the biggest difference is in Kaggle Credit, which is the largest dataset. GAN only achieves 0.3453 AUPR while the original dataset has around 0.7020. Anyway thank you for your great work!

---

> > > ### Author Response · Authors · 2018-11-25
> > > **RE: Surprising differences in Kaggle Credit**
> > >
> > >
> > > Answer: Note that the Kaggle Credit dataset is highly unbalanced (only 0.2% are positive labels). Therefore, AUPRC can be highly variable in this setting.

---

### Public Comment · ~Lovedeep_Gondara1 · 2018-11-08
**Some questions to authors**

It is an interesting work and my apologies for posting the questions at this stage. I just recently came across this work.

I see DPGAN performing reasonably well with smaller epsilon(<=1) in most cases. In original DPGAN paper however, the authors only investigated epsilon >=9 and still found it performing not so good. Especially on structured data (MIMIC-III), where they had to use epsilon >> 10 to get reasonable results. How does this implementation achieves these results? Even if one can argue the datasets are different, but still this is a consistent and significant improvement.

I really think some results on visual datasets (MNIST,LSUN) and datasets that have been previously used (such as MIMIC-III) might help shed some light on this peculiar phenomenon. It will also solidify the proposed method on reasonably high dimensional inputs. Maximum dimensionality in current paper is 35 (dataset in appendix).

Paper states that the code for DPGAN is used from “https://github.com/illidanlab”. I cannot find any DPGAN implementation in that repository. I also struggle to understand the “pre-training” of student discriminator. I think most of my above questions can be answered if authors share the implementation details of PATE-GAN.

Also, what are the sample sizes of generated synthetic data used for evaluation? I think providing a class(label) distribution in generated data would be helpful as well. Another side-point, in an imbalanced dataset such as Kaggle credit card fraud detection dataset, AUROC might not  be the most informative measure for evaluation.

---

> ### Author Response · Authors · 2018-11-14
> **RE: Some questions to authors**
>
> Thank you for your comments.
>
> A1: First, as can be seen in Figure 1 in the manuscript, the performance of DPGAN significantly decreases when epsilon is smaller than 1. You can also check this in Table 2 in the manuscript.
> Moreover, in figure 3 in DPGAN the authors investigated epsilon = 11.5, 3.2, 0.96, 0.72, and show that for digits 0 and 1, and digits 4 and 5 that their accuracy for epsilon = 0.96 is above 0.8. We believe this to be consistent with our findings. The metric they use on MIMIC is the dimension-wise prediction (DWP) which is not comparable with the metrics we use.
>
> A2: Thank you for your suggestion. We have obtained results on two higher-dimensional UCI datasets (UCI ISOLET dataset (dimensions: 617, no of samples: 7797, task: classify consonant vs vowel) and UCI Epileptic Seizure Recognition dataset (dimensions: 179, no of samples: 11500, task: classify seizure activity)) which will be included in the revised manuscript.
>
> (1) UCI ISOLET Dataset
> --------------------------------------------------------------------------------------------------
> (epsilon, delta) = (10, 10^-5) |      GAN     |     PATE-GAN     |     DPGAN    |
> --------------------------------------------------------------------------------------------------
>                AUROC                       |     0.817     |        0.769          |       0.739      |
>                AUPRC                       |     0.556     |        0.473          |       0.383      |
> --------------------------------------------------------------------------------------------------
>
> (2) UCI Epileptic Seizure Recognition Dataset
> --------------------------------------------------------------------------------------------------
> (epsilon, delta) = (10, 10^-5) |      GAN     |     PATE-GAN     |     DPGAN    |
> --------------------------------------------------------------------------------------------------
>                AUROC                       |     0.917     |        0.872          |       0.819      |
>                AUPRC                       |     0.813      |        0.766          |       0.720      |
> --------------------------------------------------------------------------------------------------
>
> A3: The implementation used the code that was published in https://github.com/illidanlab/dpgan (which was the source cited by the original DPGAN paper). Currently the DPGAN authors may be revising the code; it is not currently accessible (we do not know why). You can directly ask the DPGAN authors to share the code, but we assure you we used the code provided there.
>
> A4: We are not sure what you mean by “pre-training”. In the PATE-GAN framework, we do not “pre-train” the student discriminator. Upon an acceptance decision, we will publish our code, though we do not wish to do so until then to preserve our anonymity.
>
> A5: The sample sizes and the label distribution of the generated synthetic datasets are exactly the same as the size of the training datasets. The sample size and the label distribution of the training datasets can be found in the page 12 in the manuscript. Furthermore, we were aware that AUROC is not sufficient for imbalanced data and thus, we included AUPRC to address this problem.

---

> > ### Public Comment · ~Lovedeep_Gondara1 · 2018-11-15
> > **Further concerns**
> >
> > I thank the authors for the detailed point-wise response.
> >
> > A1.
> > I see the decline, but it still is within 4-8% of PATE-GAN and the difference surprisingly decreases with lower epsilon with DPGAN maintaining AUROC of >0.6 at the minimum epsilon. I would have expected DPGAN to be closer to random guessing at that point (based on the noise levels required for that level of privacy and DPGAN results from the manuscript). Same is true for Table 2 up to epsilon 0.1. DPGAN figure 3 doesn't really show the full picture. Binary comparison of 0,1; 2,3 and 4,5 is far from covering generator's full domain(more satisfactory would be the classification accuracy on all 10 generated digits in a single model).
> >
> > A2.
> > Thank you for providing these results. I am concerned about the epsilon used here (epsilon=10). At this level, differential privacy effectively doesn't provide much of a privacy protection. One can see the significant difference exp(10) makes in distinguishing two answers. Are the results available for epsilon<1?
> >
> > A3.
> > I have followed the source before without any luck and have emailed the authors as well in recent past(author's response was that they don't plan to release the code yet.).
> >
> > A4.
> > I understand, I was referring to the pre-training of student where Supp(P_U), Supp(P_G) is involved.
> >
> >
> > I still do think some visual results will be really helpful at different epsilon values(such as figure 1 in DPSGD).

---

> > > ### Author Response · Authors · 2018-11-25
> > > **RE: Further concerns**
> > >
> > >
> > > A1: In high-dimensional settings (such as in the UCI ISOLET dataset), the performance of DPGAN with (epsilon, delta) = (1,10^-5) is in fact close to random guessing (AUROC < 0.6). Our task in the datasets we perform experiments on is binary classification, and therefore aligns with the DPGAN two-digit classification (rather than all 10 generated digits). Again, we feel our results for DPGAN are consistent with the findings in DPGAN.
> > >
> > > A2: We have provided results for epsilon=1 (with delta = 10^-5) for both UCI ISOLET dataset and UCI Epileptic Seizure Recognition dataset in the revised manuscript.
> > >
> > > A3 & A4: Okay.
> > >
> > > A5: We feel that visual results would be qualitative at best, and do not allow for a meaningful comparison between methods. By using quantitative measure of performance we are able to directly, and fairly, compare PATEGAN and DPGAN.

---

### Public Comment · ~Yunhui_Long1 · 2019-05-13
**Sharing hyper-parameters for reproducing the experiments**

Thank you for the great work!

We are very interested in this work and are working on reproducing the experiments since authors did not open source their code. However, we found that some experiment details are not very clear in the paper. Could you please share the following details about the experiments?

1. What are the structures and hyperparmeters for GAN benchmark? Are they the same as the structures and parameters for DP-GAN (listed in the Appendix)?

2. What are the training epochs for GAN, DP-GAN, and PATE-GAN respectively?

3. What are the hyper-parameters used in the 12 predictive models? Are they all default values in sklearn?

4. Have you resampled the synthetic data to guarantee that "the label distribution of the generated synthetic datasets are exactly the same as the size of the training datasets"? If so, is the label distribution of the private dataset assumed to be public information?

---

### Public Comment · ~Florimond_Houssiau1 · 2021-06-15
**Code release**

Thanks for the interesting work!

Would it be possible for the authors to release their code, as they stated they would do in response to one of the reviewers?

Many thanks.

---

### Meta-Review · Area_Chair1 · 2018-12-17
**Advances in differentially-private data generation**

**Confidence:** 4
**Recommendation:** Accept (Poster)

**Metareview:**

This paper improves upon the PATE-GAN framework for differentially-private synthetic data generation. They eliminate the need for public data samples for training the GAN, by providing a distribution which can be sampled from instead.

The authors were unanimous in their vote to accept.